# Otological Features of Patients with Musculocontractural Ehlers–Danlos Syndrome Caused by Pathogenic Variants in *CHST14* (mcEDS-*CHST14*)

**DOI:** 10.3390/genes14071350

**Published:** 2023-06-27

**Authors:** Masayuki Kawakita, Satoshi Iwasaki, Hideaki Moteki, Shin-ya Nishio, Tomoki Kosho, Shin-ichi Usami

**Affiliations:** 1Asama General Hospital, Saku 385-8558, Japan; rmchampionmkred@gmail.com; 2Department of Hearing Implant Sciences, Shinshu University School of Medicine, Matsumoto 390-8621, Japan; iwasakis@iuhw.ac.jp (S.I.); hideaki.moteki@gmail.com (H.M.); nishio@shinshu-u.ac.jp (S.-y.N.); usami@shinshu-u.ac.jp (S.-i.U.); 3Department of Otolaryngology, International University of Health and Welfare, Mita Hospital, Tokyo 108-8329, Japan; 4Department of Otolaryngology, Aizawa Hospital, Matsumoto 390-8621, Japan; 5Department of Medical Genetics, Shinshu University School of Medicine, Matsumoto 390-8621, Japan; 6Center for Medical Genetics, Shinshu University Hospital, Matsumoto 390-8621, Japan; 7Division of Clinical Sequencing, Shinshu University School of Medicine, Matsumoto 390-8621, Japan; 8Research Center for Supports to Advanced Science, Shinshu University, Matsumoto 390-8621, Japan

**Keywords:** musculocontractural Ehlers–Danlos syndrome, mcEDS-*CHST14*, hearing loss, sensorineural, no DPOAE response

## Abstract

Musculocontractural Ehlers–Danlos syndrome (EDS) caused by pathogenic variants in *CHST14* (mcEDS-*CHST14*) is a subtype of EDS characterized by multisystem malformations and progressive fragility-related manifestations. A recent international collaborative study showed that 55% of mcEDS-*CHST14* patients had hearing loss (HL), more commonly of the high-frequency type. Here, we report the first systemic investigation of the otological features of patients with this disorder based on the world’s largest cohort at Shinshu University Hospital. Nine patients [18 ears; four male and five female patients; mean age, 18 years old (range, 10–28)] underwent comprehensive otological evaluation: audiogram, distortion product otoacoustic emission (DPOAE) test, and tympanometry. The audiogram, available in all 18 ears, showed HL in eight patients (8/9, 89%) and in 14 ears (14/18, 78%): bilateral in six patients (6/9, 67%) and unilateral in two (2/9, 22%); mild in eight ears (8/18, 44%) and moderate in six (6/18, 33%); and high-frequency HL in five (5/18, 28%) and low-frequency HL in five (5/18, 28%). An air-bone gap was detected in one ear (1/18, 6%). DPOAE was available in 13 ears, with the presence of a response in five (5/13, 38%) and the absence in eight (8/13, 62%), including in three ears of normal hearing. Tympanometry results were available in 12 ears: Ad type in nine (9/12, 75%) and As type in one (1/12, 8.3%). Patients with mcEDS-*CHST14* had a high prevalence of HL, typically sensorineural and bilateral, with mild to moderate severity, of high-frequency or low-frequency type, and sometimes with no DPOAE response. The pathophysiology underlying HL might be complex, presumably related to alterations of the tectorial membrane and/or the basilar membrane of Corti associated with disorganized collagen fibril networks. Regular and careful check-ups of hearing using multiple modalities are recommended for mcEDS-*CHST14* patients.

## 1. Introduction

Ehlers–Danlos syndrome (EDS) is a clinically and genetically heterogeneous connective tissue disorder characterized by skin hyperextensibility, joint hypermobility, and tissue fragility. The 2017 International Classification of EDS [1] and a subsequent review [2] recognized a total of 14 EDS subtypes: classical EDS, classical-like EDS, cardiac valvular EDS, vascular EDS, hypermobile EDS, arthrochalasia EDS, dermatosparaxis EDS, kyphoscoliotic EDS, brittle cornea syndrome, spondylodysplastic EDS, musculocontractural EDS, myopathic EDS, periodontal EDS, and classical-like EDS type 2. 

Musculocontractural EDS (mcEDS) is a hereditary connective tissue disorder with autosomal recessive inheritance, associated with generalized depletion of dermatan sulfate (DS), which is caused by defective activity of dermatan 4-*O*-sulfotransferase 1 (D4ST1) or dermatan sulfate epimerase (DSE) through biallelic loss-of-function variants in the carbohydrate sulfotransferase gene (*CHST14*) (mcEDS-*CHST14*) (MIM#601776) or in the gene encoding DSE (*DSE*) (mcEDS-*DSE*) (MIM#615539), respectively [3]. McEDS-*CHST14* was identified as a rare arthrogryposis syndrome named “adducted thumb-clubfoot syndrome” by Dündar, Janecke, and their colleagues in 2009 [4], a new subtype of EDS named “Kosho type” by Miyake, Kosho, and their colleagues in 2010 [5], and a subtype of kyphoscoliotic EDS by Malfait and her colleagues in 2010 [6]. McEDS-*DSE* was identified by Janecke and his colleagues in 2013 [7].

The clinical characteristics of mcEDS-*CHST14* include multisystem malformations (craniofacial features, congenital multiple contractures, visceral and ocular malformations) and progressive fragility-related manifestations (skin hyperextensibility and fragility, recurrent dislocations, spinal and foot deformities, large subcutaneous hematomas) [3,4,5,6,7,8,9,10]. Major criteria comprise (1) congenital multiple contractures, characteristically adduction-flexion contractures and/or talipes equinovarus (clubfoot); (2) characteristic craniofacial features, which are evident at birth or in early infancy; and (3) characteristic cutaneous features including skin hyperextensibility, easy bruisability, skin fragility with atrophic scars, and increased palmar wrinkling. Minor criteria comprise recurrent/chronic dislocations, pectus deformities (flat, excavated), spinal deformities (scoliosis, kyphoscoliosis), peculiar fingers (tapering, slender, cylindrical), progressive talipes deformities (valgus, planus, cavum), large subcutaneous hematomas, chronic constipation, colonic diverticula, pneumothorax/pneumohemothorax, nephrolithiasis/cystolithiasis, hydronephrosis, cryptorchidism in males, strabismus, refractive errors (myopia, astigmatism), and glaucoma/elevated intraocular pressure [1]. Minimum criteria suggestive for mcEDS and proposing confirmatory molecular testing include: at birth or in early childhood, congenital multiple contractures and characteristic craniofacial features; in adolescence and in adulthood, congenital multiple contractures and characteristic cutaneous features [1].

The skin features (hyperextensibility, fragility) are supposedly attributable to structural changes in the glycosaminoglycan (GAG) sidechains of decorin, a major DS-proteoglycan in the skin that contributes to collagen fibril assembly. Specifically, affected GAG chains are linear and stretched from the outer surface of collagen fibrils to adjacent fibrils, whereas normal GAG chains are curved and maintain close contact with attached collagen fibrils [9,11]. These structural changes are considered to occur through changes in the composition of the GAG chains from mostly DS to exclusively chondroitin sulfate (CS), associated with the generalized depletion of DS [9,11].

A recent international collaborative study retrospectively showed that 22/40 patients (55%) with mcEDS-*CHST14* had hearing loss; high-frequency loss was more common (n = 10) than low-frequency loss (n = 2) [10]. Here, we report the first systemic investigation of the otological features of patients with mcEDS-*CHST14*, including the patterns and severity, based on the world’s largest cohort at Shinshu University Hospital.

## 2. Materials and Methods

Nine patients [18 ears from four male and five female patients; mean age, 18 years old (range, 10–28 years old)] with mcEDS-*CHST14* who underwent comprehensive otological evaluation at Shinshu University Hospital were recruited (Table 1). Six patients had compound heterozygous variants in *CHST14*, and three had variants homozygously. The most common variant “p.[Pro281Leu]” [6] were shared by seven patients. Cardinal features in mcEDS-*CHST14* were observed in all patients, including craniofacial characteristics, congenital multiple contractures, skin hyperextensibility and fragility, recurrent dislocations, spinal and foot deformities, and large subcutaneous hematomas.

Subjective symptoms and the results of the pure tone hearing test (audiogram), distortion product otoacoustic emission (DPOAE) test, and tympanometry test were collected through electronic medical records. The air-bone gap (AB gap) was described as positive in an ear if the gap exceeded 20 dB and measured at 250 Hz (low frequency) and at 4000 Hz (high frequency). The DPOAE test was evaluated in the range of 1 kHz–6250 Hz.

HL was diagnosed using an air-conduction hearing level >25 dB as follows: mild, 26–40; moderate, 41–55; moderately severe, 56–70; severe, 71–90; and profound, >91. Low-frequency HL was defined as HL at 125, 250, or 500 Hz; middle-frequency HL was defined as HL at 1000 or 2000 Hz; and high-frequency HL was defined as HL at 4000 or 8000 Hz. Ad type was defined as a peak value of compliance ≥1.0 mL as determined by tympanometry.

## 3. Results

Otologocal symptoms of nine patients are summarized in Table 2. Only a male patient was aware of hearing loss (HL), whereas five patients were not aware of HL. Only a male patient who was aware of HL had tinnitus, whereas four patients mentioned no tinnitus. Two patients mentioned no dizziness.

Otological findings using audiogram, DPOAE, and tympanometry test in the 18 ears from nine patients are summarized in Table 3 and Figure 1. The audiogram showed HL in eight patients (8/9, 89%) [bilateral HL in six patients (6/9, 67%) and unilateral HL in two (2/9, 22%)] and no HL in one (1/9, 11%). HL was found in 14 ears (14/18, 78%): mild HL in eight (8/18, 44%) and moderate HL in six (6/18, 33%). HL was of the high-frequency type in five ears (5/18, 28%), middle-frequency type in one (1/18, 6%), low-frequency type in five ears (5/18, 28%), high- and middle-frequency type in one (1/18, 6%), high- and low-frequency type in one (1/18, 6%), and all-frequency type in one (6%). An AB gap was detected in one ear (1/18, 6%).

The DPOAE test, evaluated only at 1000 Hz and excluding five ears in which substantial noise was recorded, showed the presence of a response in five ears (5/13, 38%). Among eight ears (8/13, 62%) with the absence of a response, the audiogram showed high-frequency moderate HL in three, low-frequency moderate HL in one, low and high-frequency mild HL in one, and normal hearing in three.

The results of tympanometry, available in 12 ears, showed Ad type in nine ears (9/12, 75%), As type in one ear (1/12, 8.3%), and no abnormalities (A) in two ears (2/12, 17%).

## 4. Discussion

Otological features were investigated in this first detailed and comprehensive series on patients with mcEDS-*CHST14*. Patients with mcEDS-*CHST14* frequently manifest HL (67% of patients, 78% of ears) typically bilaterally, at mild to moderate severity, and of high-frequency or low-frequency type, with no AB gap indicating sensorineural HL. No DPOAE response was observed in eight ears, including three of normal hearing. Patients typically showed Ad-type tympanometry. Because all the modalities were performed in the clinical setting, they were performed not by multiple evaluators but by experienced clinical technicians, which might affect the accuracy of the results. However, all the modalities are established and used in a daily ontological practice, so differences among evaluators would be minimum.

The identified prevalence of HL in patients with mcEDS-*CHST14* was comparable to the recent finding in an international collaborative study (55% of patients) [10]. The predominant origin of HL was sensorineural in mcEDS-*CHST14*, whereas HL was equally of conductive and sensorineural origin in the largest series of audiological outcomes in EDS, comprising various subtypes [12]. Therefore, an initial otological investigation after the diagnosis and subsequent check-ups is recommended for patients with mcEDS.

The lack of a DPOAE response in eight ears, including three of normal hearing, is an interesting finding in this study. The DPOAE test is an established method for detecting otoacoustic emissions (OAE), which reflects the function of the outer hair cells [13], which are attached to the tectorial membrane via their stereocilia [14]. OAE and cochlear status are closely related: damaged and missing outer hair cells result in changed, typically diminished, emissions. It is, therefore, widely used for hearing loss screening and diagnostics in infants, children, and adults [13].

Because collagen, primarily of type II, is a predominant protein in the tectorial membrane [14], structural and subsequently functional alterations might occur in the tectorial membrane of patients with mcEDS-*CHST14* via disorganized collagen fibril networks, as observed in the affected skin [9,11] as well as in the *Chst14^−/−^* mice [15]. However, this assumption would not comprehensively explain the pathophysiology of HL in mcEDS-*CHST14* because the presence or absence of a DPOAE response did not correlate with hearing conditions in the current cohort. The basilar membrane of Corti also contains large amounts of collagen, primarily of type II [16], and its structural and subsequently functional alterations might be related to HL in this cohort. The complex pathophysiology was thus estimated to occur through alterations of the tectorial membrane and/or the basilar membrane of Corti associated with disorganized collagen fibril networks.

The Ad-type tympanometry observed in three-quarters of the affected ears suggests a hypermobile tympanic membrane with normal middle-ear pressure along with increased static compliance. The tympanic membrane consists of diverse collagen types: the lamina propria of the pars tensa made up of type II collagen and the lamina propria of the pars flaccida made up of type I collagen. Furthermore, during the healing process or infection, type I and type III collagens are present [14]. The supposed hypermobility in the tympanic membrane would be attributable to the disorganization of collagen fibril networks mediated by structural changes (dermatan sulfate to chondroitin sulfate) in the GAG sidechains of decorin, which has been demonstrated to play a critical role in the assembly of collagen fibrils in the skin [9,11].

## 5. Conclusions

This first systemic otological investigation of mcEDS-*CHST14* demonstrated a high prevalence of HL, typically sensorineural and bilateral, of mild to moderate severity, and of high-frequency or low-frequency type, and sometimes with no DPOAE response regardless of the hearing condition. Regular and careful check-ups of hearing using multiple modalities are recommended in patients with mcEDS-*CHST14*.

## Figures and Tables

**Figure 1 genes-14-01350-f001:**
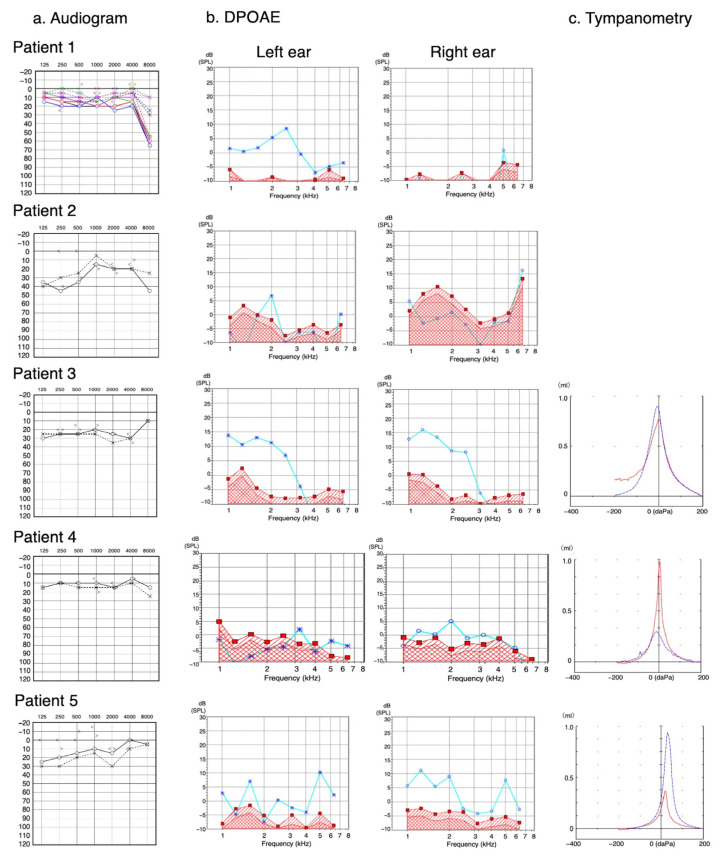
Images of audiogram, distortion product otoacoustic emission (DPOAE), and tympanometry in each patient. (**a**) Audiogram. The horizontal axis and vertical axis indicate frequency (Hz) and hearing threshold (dB), respectively. At the air-conduction testing, “×” and “○” indicate “left” and “right”, respectively; and the same line color indicates the result obtained at the same testing. At the bone-conduction testing, “>” and “<” indicate “left” and “right”, respectively. (**b**) DPOAE. The horizontal axis and vertical axis indicate frequency (Hz) and hearing threshold (dB), respectively. “×” and “○” indicate “left” and “right”, respectively. Light blue lines indicate the results of DPOAE, and areas under the red lines indicate the background noise. (**c**) Tympanometry. The horizontal axis and vertical axis indicate external ear canal pressure (daPa) and external ear canal volume (mL), respectively. Blue curve and red curve indicate left and right ears, respectively.

**Table 1 genes-14-01350-t001:** Clinical and molecular characteristics of patients with mcEDS-*CHST14* in this cohort.

Patient	Sex	Variant (NM_130468.4)	Protein Alteration (NP_569735.1)	Major Complications
1	F	c.[842C>T];[878A>G]	p.[Pro281Leu];[Tyr293Cys]	Dislocation, kyphoscoliosis, large subcutaneous hematoma, fistula, massive skin necrosis, gastric ulcer
2	F	c.[842C>T];[878A>G]	p.[Pro281Leu];[Tyr293Cys]	Dislocation, large subcutaneous hematoma
3	M	c.[2_10del];[2_10del]	p.[?];[?]	Kyphoscoliosis, large subcutaneous hematoma, retinal detachment
4	F	c.[842C>T];[842C>T]	p.[Pro281Leu];[Pro281Leu]	Dislocation, kyphoscoliosis, foot deformity, ureteral stone, large subcutaneous hematoma
5	F	c.[2_10del];[676_682delinsGCTATGGGGCT]	p.[?];[Lys226Alafs*16]	Dislocation, meniscus tear, large subcutaneous hematoma
6	M	c.[842C>T];[878A>G]	p.[Pro281Leu];[Tyr293Cys]	Foot deformity
7	M	c.[626T>C];[842C>T]	p.[Phe209Ser];[Pro281Leu]	Skin laceration, large subcutaneous hematoma
8	F	c.[626T>C];[842C>T]	p.[Phe209Ser];[Pro281Leu]	Dislocation, kyphoscoliosis, large subcutaneous hematoma
9	M	c.[842C>T];[842C>T]	p.[Pro281Leu];[Pro281Leu]	Dislocation, skin laceration, large subcutaneous hematoma, colon diverticulitis

F, female; M, male. Variants are described according to the variant nomenclature of the Human Genome Variation Society (https://varnomen.hgvs.org).

**Table 2 genes-14-01350-t002:** Otological symptoms of patients with mcEDS-*CHST14* in this cohort.

Patient	Age at the Time of Otological Evaluation (Years)	Awareness of Hearing Loss	Tinnitus	Dizziness
1	25	No	No	No
2	13	No	N/A	No
3	18	N/A	N/A	N/A
4	18	N/A	N/A	N/A
5	17	No	No	N/A
6	10	No	No	N/A
7	14	Yes	Yes	N/A
8	28	No	No	N/A
9	19	N/A	N/A	N/A

N/A, data not available.

**Table 3 genes-14-01350-t003:** Otological findings of patients with mcEDS-*CHST14* in this cohort.

Patient/Ear	Audiogram	AB Gap	DPOAE	Tympanogram
1/R	Moderate (high)	Negative	Absence	N/A
1/L	Normal	Negative	Presence	N/A
2/R	Moderate (low)	Positive (L)	N/A ^#^	N/A
2/L	Moderate (low)	Negative	Absence	N/A
3/R	Mild (high)	Negative	Presence	Ad
3/L	Mild (high)	Negative	Presence	A
4/R	Normal	Negative	Absence	Ad
4/L	Normal	Negative	Absence	As
5/R	Mild (low)	Negative	Presence	Ad
5/L	Mild (low)	Negative	Presence	Ad
6/R	Mild (low)	Negative	N/A ^#^	N/A
6/L	Moderate (all)	Negative	N/A ^#^	N/A
7/R	Mild (moderate, high)	Negative	N/A ^#^	Ad
7/L	Mild (moderate)	Negative	N/A ^#^	A
8/R	Normal	Negative	Absence	Ad
8/L	Mild (low, high)	Negative	Absence	Ad
9/R	Moderate (high)	Negative	Absence	Ad
9/L	Moderate (high)	Negative	Absence	Ad

DPOAE, distortion product otoacoustic emission; L, left side; R, right side; N/A, data not available; ^#^, noise.

## Data Availability

The data that support the findings of this study are available from the corresponding author upon reasonable request.

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
