# Peer review of "Otological Features of Patients with Musculocontractural Ehlers–Danlos Syndrome Caused by Pathogenic Variants in CHST14 (mcEDS-CHST14)"

_genes, 2023, doi:10.3390/genes14071350_

Round 1
Reviewer 1 Report
The results of audiological investigation in pathogenic variants of gene CHST14 are presented. Paper definitely is of scientific and practical interest.
Comments:
In the Material and Methods section is no information on the pathogenic variants revealed in patients. This information is very important for this type of papers and for readers of the Journal.
It is not clear had patients any complaints or this investigation was planned with the idea to reveal the hearing loss in that group of patients?
The description of the distortion product otoacoustic emission should be clarified.
There are no data on the severity of other symptoms, it is not clear what pathogenic variants were revealed (confirmed). This information would be of great importance.
The recommendation to investigate hearing is given, but it is not clear if all patients had complaints on hearing loss or not?
The presented paper is descriptive so targeted percentages are confusing, for example, in abstract - line 26-32, in results – line 88-102 etc.
Author Response
Dear Reviewer 1
The results of audiological investigation in pathogenic variants of gene CHST14 are presented. Paper definitely is of scientific and practical interest.
Response: Thank you so much for this high evaluation.
Comments:
In the Material and Methods section is no information on the pathogenic variants revealed in patients. This information is very important for this type of papers and for readers of the Journal.
Response: Thank you. We have included variant information in Table 1.
It is not clear had patients any complaints or this investigation was planned with the idea to reveal the hearing loss in that group of patients?
Response: Thank you. This investigation was planned and prospectively conducted based on a literature-based risk of hearing loss in patients with mcEDS.
The description of the distortion product otoacoustic emission should be clarified.
Response: Thank you. We have clarified the description of DPOAE in the 3rd paragraph of the discussion as follows, adding one literature:
The lack of a DPOAE response in four ears, including two of normal hearing, is an interesting finding in this study. DPOAE is an established method to detect otoacoustic emissions (OAE), which reflects function of the outer hair cells [11], which are attached to the tectorial membrane via their stereocilia [12]. OAE and cochlear status are closely related: damaged and missing outer hair cells result in changed, typically diminished, emissions. It is, therefore, widely used for hearing loss screening and diagnostics in infants, children, and adults [13].
Reavis, K. M., McMillan, G. P., Dille, M. F., Konrad-Martin, D. Meta-Analysis of Distortion Product Otoacoustic Emission Retest Variability for Serial Monitoring of Cochlear Function in Adults. Ear and hearing. 2015, 36(5), e251–e260; DOI:10.1097/AUD.0000000000000176.
There are no data on the severity of other symptoms, it is not clear what pathogenic variants were revealed (confirmed). This information would be of great importance.
Response: Thank you. We have described variant information as well as major complications in Table 1.
The recommendation to investigate hearing is given, but it is not clear if all patients had complaints on hearing loss or not?
Response: Thank you. Only one patient was aware of hearing loss, but in those who were not aware of hearing loss some though not severe hearing problems existed. Because the clinical evidence of this newly established subtype of EDS is still limited, the current data could be sufficient to recommend regular otological check-ups to patients with mcEDS.
The presented paper is descriptive so targeted percentages are confusing, for example, in abstract - line 26-32, in results – line 88-102 etc.
Response: Thank you. According to this comments, we have clarified what these percentages meant in the abstract and the result.
I only have a few minor comments:
Line 73 : Please replace "She" with The clinical course of the disease...
Response: I am so sorry, I could not find the part.
Sincerely yours,
Tomoki Kosho, M.D.
Reviewer 2 Report
in the manuscript authors present otological features in patients with MEDS, these are some of my comments
In the phrase ".....The results of tympanometry, available in 12 ears...."please define why 12
Who did perform the studies ? Were there multiple evaluators?, in this case please mention Kappa results.
The discussion must be scientifically enriched by comparing its results with previous reports
The main weakness of the study is mentioning variants in CHST14 without presenting evidence in this regard. The study must show molecular evidence of these variants with methodological support and not just a brief description of the clinical findings.
no comments
Author Response
Dear Reviewer 2
in the manuscript authors present otological features in patients with MEDS, these are some of my comments
Response: Thank you.
In the phrase ".....The results of tympanometry, available in 12 ears...."please define why 12
Response: Thank you. In six ears, tympanometry data was not appropriate to be interpreted accurately.
Who did perform the studies ? Were there multiple evaluators?, in this case please mention Kappa results.
Response: Thank you. All these procedures were performed in the clinical setting, where not multiple evaluators but experienced clinical laboratory technicians did them routinely. However, this point could be important, so relevant description has been added in the 1st paragraph of the discussion as follows:
“Because all the modalities were performed in the clinical setting, they were performed not by multiple evaluators but by experienced clinical technicians, which might affect accuracy of the results. However, all the modalities are established and used in a daily ontological practice, so differences among evaluators would be minimum.”
The discussion must be scientifically enriched by comparing its results with previous reports
Response: Thank you. However,
The main weakness of the study is mentioning variants in CHST14 without presenting evidence in this regard. The study must show molecular evidence of these variants with methodological support and not just a brief description of the clinical findings.
Response: Thank you. We have added variants detected in each individual in Table 1.
Sincerely yours,
Tomoki Kosho, M.D.
Round 2
Reviewer 2 Report
PLease, only include the electropherogram (figure) of the variant
No comments
Author Response
Dear Reviewer 2
Thank you for your comment regarding the inclusion of a figure showing electropherograms of each variant. However, all these cases were presented in our recent large-scale study (Minatogawa, Kosho et al., J Med Genet 59(9):865-877,2022; PMID: 34815299) and the variants were established in the paper. So it would not be necessary for electropherograms to be disclosed in this paper focusing on otologcial features of this cohort.
Sincerely yours,
Tomoki Kosho, M.D.